# Ultra-Processed Food Consumption as a Risk Factor for Gastrointestinal Cancer and Other Causes of Mortality in Southern Italy: A Competing Risk Approach

**DOI:** 10.3390/nu16131994

**Published:** 2024-06-23

**Authors:** Angelo Campanella, Rossella Tatoli, Caterina Bonfiglio, Rossella Donghia, Francesco Cuccaro, Gianluigi Giannelli

**Affiliations:** 1National Institute of Gastroenterology—IRCCS “Saverio de Bellis”, 70013 Castellana Grotte, Italy; rossella.tatoli@irccsdebellis.it (R.T.); catia.bonfiglio@irccsdebellis.it (C.B.); rossella.donghia@irccsdebellis.it (R.D.); gianluigi.giannelli@irccsdebellis.it (G.G.); 2Local Health Unit—Barletta-Andria-Trani, 76121 Barletta, Italy; francescocuccaroepi@gmail.com

**Keywords:** ultra-processed foods, gastrointestinal cancer, mortality, competing risk

## Abstract

Background: Ultra-Processed Foods (UPFs) are increasingly consumed worldwide, even in regions with strong dietary traditions like the Mediterranean and can play a crucial role in the development of chronic diseases, including cancer. This population-based prospective cohort study investigates the association between UPF consumption and gastrointestinal cancers and other causes of mortality in Southern Italy. Methods: Data were collected from 4870 participants in the MICOL and NUTRIHEP cohorts. The EPIC questionnaire was used to elicit information on food and drink consumption and UPFs were categorized by degree of processing according to the NOVA classification. Cox proportional hazards regression and competing risk models were employed for statistical analysis. Results: UPF consumption was positively associated with all-cause mortality: participants in the 3rd UFP quartile, as compared to the lowest, had a 27% higher risk of death (SHR 1.27 95% CI, 1.03; 1.57), while in the highest quartile as compared to the lowest, the risk was 34% higher (SHR 1.34 95% CI, 1.00; 1.79). Higher UPFs intake was also correlated with an increased gastrointestinal cancers mortality risk, especially the 2nd (SHR 1.65, 95% CI: 1.01; 2.71) and 4th quartile (SHR 3.14 95% CI: 1.56; 6.32), with a dose-dependent effect. For the other cancers, a SHR 1.61 (95% CI 1.03; 2.54) was observed for the 3rd quartile. Conclusions: Our results reinforce the link between UPF consumption and cancer risk, emphasizing the urgent need for interventions targeting dietary patterns.

## 1. Introduction

Chronic diseases are a growing health challenge today, including the most common diseases such as type 2 diabetes, cardiovascular disease and cancer [1], which are the leading causes of mortality and morbidity worldwide [2]. In economically developed countries, the prevalence of people with more than one chronic disease has increased dramatically over the past 20 years [2], and the same trend is being observed in developing countries [3]. 

Diet is a major modifiable risk factor for these pathological conditions and a significant contributor to the overall burden of disease [4], accounting for approximately 22% of noncommunicable disease deaths in the adult population [5].

Ultra-Processed Foods (UPFs) contribute 30–50% of the daily caloric intake of the world population nowadays [6], with a higher consumption in high-income countries, and increasing in middle-income countries [7,8].

The consumption of UPFs is on the rise, even in countries with a strong tradition of the Mediterranean diet [9], and this trend is attributed to factors such as affordability, convenience, and accessibility. 

According to the NOVA classification, UPFs are industrially manufactured products comprising deconstructed and modified food components recombined with various additives [6]. These foods, including industrially produced cereals and cookies, soft and sweetened drinks, are high in added sugars, vegetable oils and fats, and low in nutrients and bioactive compounds [10]. The increasing replacement of fresh or minimally processed foods with UPFs makes the diet very poor, reducing its protective effects on health [11,12].

Since the inception of the NOVA classification, the scientific community has shown a growing interest in the effects of a UPF-rich diet on health status [13].

Several studies have shown positive associations between UPF consumption and the risk of cancer [12]. A multicentric population based case-control study in Spain reported a positive association between UPF consumption and the risk of colorectal cancer [14]. A diet rich in ultra-processed foods increases the risk of gut dysbiosis [15] and the pro-inflammatory potential of the gut microbiome, promoting colon carcinogenesis [16,17,18].

The present study aims to investigate the association between UPF consumption and the risk of death from all causes and gastrointestinal cancers in a Southern Italian population. 

## 2. Materials and Methods

### 2.1. Study Population

Details of the study population have been published elsewhere [19,20]. The MICOL Study [21] is a population-based prospective cohort study randomly drawn from the electoral rolls of Castellana Grotte (≥30 years) in 1985 and followed over time with 3 follow ups: (1992–1993; 2005–2006; 2017–2019). This work refers to the second follow up (2005–2006) when a random sample of subjects (PANEL Study) aged 30–50 years was added to the initial cohort to compensate for the ageing of the cohort. 

The NUTRIHEP study is a cohort extracted in 2005–2006 from the registries of general practitioners in the city of Putignano (≥18 years), based on the assumption that there were no differences in the distribution of sex and age in these lists, compared to the general population [22].

A total of 5271 eligible subjects out of 5378 (98.1% response rate) gave written informed consent to participate. Participants who had not completed the lifestyle and diet questionnaires were excluded from the present study: 110 from MICOL and 291 from NUTRIHEP (Figure 1).

All procedures were performed according to the ethical standards of the institutional research committee (National Institute of Gastroenterology, IRCCS “S. De Bellis” Research Hospital), after the ethical committee approved the MICOL Study (DDG-CE-589/2004 18 November 2004) and the NUTRIHEP Study in 2005 (DDG-CE-502/2005 20 May 2005).

The study was conducted according to the 1964 Helsinki Declaration and later amendments, and written informed consent was obtained from each participant.

### 2.2. Data Collection

The study subjects were interviewed to collect information on sociodemographic characteristics, health status, personal history and lifestyle factors including tobacco use (Never and Current), food intake, and educational level (Illiterate, Primary School, Secondary School, High School, Graduate) [23], work (Pensioners and Jobless, Managers and Professionals, Craft, Agricultural and Sales Workers, Housewives and Elementary Occupations) [24] marital status (Single, Married/Coupled, Separated/Divorced and Widow/er).

Weight was taken with the subject in underwear, standing on an electronic balance, SECA^®®^, and was approximated to the nearest 0.1 kg. Height was measured with a wall-mounted stadiometer SECA^®®^, approximated to 1 cm. Blood pressure (BP) measurement was performed following international guidelines [25]. The average of 3 BP measurements was calculated. 

Usual food intakes were estimated by administering a nation-specific validated dietary questionnaire, filled in with the support of trained nutritionists: the European Prospective Investigation on Cancer (EPIC) Food Frequency Questionnaire (FFQ), and individual nutrient intakes were derived from foods included in the dietary questionnaires through the standardized EPIC Nutrient Database [26,27].

Fasting venous blood samples were drawn, and the serum was separated into two aliquots. One aliquot was immediately stored at −80 °C. The second aliquot was used to test biochemical serum markers by standard laboratory techniques in our Central Laboratory.

### 2.3. Outcome Assessment

Participants were followed up until 31 December 2023 and the vital status or any emigrations were verified through the registry office of the municipalities of Castellana Grotte and Putignano. 

Information on causes of death from 2006 to December 2023 was extracted from the Apulian Regional Registry, using the death certificate according to WHO guidelines [28].

Causes of death were grouped as follows: Cardiovascular Disease (CVD) (ICD-10 I00-I99), Gastrointestinal Cancer (GC) (codes C14-C26), and Other Cancer (OCr) (ICD-10 C01-C013 C30-C97). Deaths from Other Causes (DOC) included the remaining ICD-10 codes. 

### 2.4. Exposure Assessment

NOVA classifies all foods and food products in four groups, based on the extent of industrial treatment to which they are subjected. In addition, NOVA considers all physical, biological and chemical methods that are used during the food production process, including the use of additives [29].

Ultra-processed foods, which fall into the category we considered as an exposure variable include carbonated soft drinks; sweet or salty packaged snacks; chocolate, candies (sweets); ice cream; mass-packaged breads and buns; margarines and other spreads; cookies, pastries, cakes, and cake mixes; breakfast cereals; ready-made cakes and pasta and pizza dishes; poultry and fish nuggets and sticks, sausages, hamburgers, hot dogs, and other reconstituted meat products; powdered and packaged soups, noodles, and “instant” desserts; and many other products [13].

We tallied up how often participants consumed ultra-processed foods each day, then grouped them into quartiles according to their daily consumption.

### 2.5. Statistical Analysis

For analytical purposes, the UPFs daily consumption was grouped into quartile categories: Q1 (<80 g/day), Q2 (80–140.9 g/day), Q3 (141–240 g/day) and Q4 (>240 g/day).

Data are presented as mean (±SD) and median (±IQR) for continuous data, and frequency (%) for categorical data. 

Time from enrolment to death, moving elsewhere or end of the study (December 31st, 2023), whichever occurred first, was the observation time.

Since age is the most important risk factor for death, we chose age at death as the time scale.

We considered 90 years of age as the maximum observation age to avoid problems related to comorbidities in the over 90s and worse coding quality of deaths occurring in very old age.

Schoenfeld residuals were calculated to test the proportional hazards assumption.

We used Cox proportional hazards regression with age as the underlying time metric to estimate hazard ratios (HRs) and 95% confidence intervals (CI) for the association between UPFSs consumption and all-cause mortality. The Cox model was adjusted for gender (F vs M), BMI categories, Alanine Aminotransferase, Triglycerides, Glucose, Smoking, Job, Marital Status, Alcohol intake, and Daily Calories. We used Q1 as a reference group. We fitted the hazard function using post-estimation tools.

For cause-specific mortality, flexible parametric survival models were run for subdistributions using a competing risk approach [30]. We estimated the Subdistribution Hazard Ratio (SHR) for the UPFs quartiles association with the risk of developing four types of competing events: CVD, Gastrointestinal Cancer, Other Cancer and death from other causes. Using post-estimation tools, we fitted the cause-specific Cumulative Incidence Function.

We constructed two models: Model 1 included age (timescale) and UPF consumption, divided into quartiles, while Model 2 added gender (F vs M), BMI categories, Alanine Aminotransferase, Triglycerides, Glucose, Smoking habit, Work, Marital status, Alcohol intake, and Daily Calories as potential confounders in Model 1.

All statistical analyses were performed using Stata, Statistical Software version 18.0 (StataCorp, College Station, TX, USA).

## 3. Results

A total of 935 (19.2%) subjects died during the observation time (27,562.302 person-years with an incidence rate of 33.9 for 1000 person-years), 271 (29.5%) from cardiovascular diseases, 268 (28.7%) from neoplastic diseases, 105 (11.2%) from gastrointestinal cancers (among which 22 malignant tumor of the colon, 34 Malignant tumor of the liver and intrahepatic bile ducts, 20 Malignant tumor of the pancreas) and the remaining 396 (42.3%) from other causes. The distribution of the 105 gastrointestinal cancers is shown in Appendix A.

The main characteristics of the 4870 subjects, classified according to quartiles of UPFs daily consumption, are shown in Table 1.

The mean age of the 3935 participants still alive was 64.8 years (±13.5), while the average age at death was 80.8 years (±10.8), 82.37 (±10.5) for women and 79.7 (±10.9) for men. 

A very interesting downward trend is the average age of participants. In Q1 we observed an average age of 63.36 (±12.79), in Q2 54.32 (±13.73), 48.04 (±13.54) in Q3 and 40.22 (±13.53) in Q4.

The results of mortality hazard ratios (HR) and Subdistribution Hazard Ratios (SHR) according to the UPFs quartiles consumption are shown in Table 2.

When analyzing UPF consumption (Table 2 Model 2), we find that the participants with the 3rd quartile of UPFs compared to the lowest had a 27% higher risk of death, from all causes (SHR 1.27 95% CI, 1.03; 1.57), while in the highest quartile compared to the lowest, the risk was 34% higher (SHR 1.34 95% CI, 1.00; 1.79).

Details by cause are shown in Table 2, Model 2: for Gastrointestinal Cancer, a statistically significant effect was found for Q2 (SHR 1.65, 95% CI: 1.01; 2.71) and highly significant for Q4 (SHR 3.14 95% CI: 1.56; 6.32) when compared with the lowest quartile, while for other cancers, a SHR 1.61 (95% CI 1.03; 2.54) was observed for Q3.

No statistical significance was found for CVD and OCD in both the first model and the multivariate model.

For all causes of mortality, a clear trend of risk curves is observed in Figure 2, with the lowest risk related to the first quartile.

The risk change of all-cause mortality due to different consumption of processed foods is observed from age 73 (Figure 2), compared with the risk change for Gastrointestinal Cancers (Figure 3), found as early as age 60, a full 13 years earlier.

At 80 years of age, the probability of death from Gastrointestinal Cancer was 2% for those with UPF consumption <80 g and 5% for those with UPF consumption >240 g/day. At age 90, the probability of death from Gastrointestinal Cancer is 5% for those with UPF consumption <80 g and 13% for those with UPF consumption >240 g/day.

Appendix A shows the cumulative incidence of other causes of death due to Other Cancer by first, second and third quartiles. 

## 4. Discussion

Our findings suggest that the growing consumption of UPFs may be a contributing factor to increasing mortality for all causes.

In particular, we observed a positive, dose-dependent association between the intake of these foods and the incidence of gastrointestinal cancers. These results confirm and extend previous evidence on this topic and underline the importance of considering UPFs as a potential risk factor for cancer [31].

However, recent years have witnessed an escalating consumption of UPFs in Mediterranean countries, driven by factors like lower cost, convenience, and ease of availability, especially for the younger generation [29].

Observational data from our cohort confirm that younger subjects were exposed to higher UPF consumption. Our analysis showed an exponentially increased risk of Gastrointestinal Cancer mortality after the age of sixty for people who consumed UPFS > 240 g/day. 

These foods often provide high-calorie density, a high percentage of added fat and sugar, low fiber, and determine a low degree of satiety. They are often offered in large portions and are designed to be eaten as snacks [32,33] and not as main meals, which potentially increases consumption and thus contributes to metabolic imbalances and an increased incidence of metabolic-related diseases.

UPFs have been associated with excess weight (i.e., obesity and high BMI) and central adiposity in several observational studies [34,35]. 

In a recent umbrella review [36], experts delved into the intake of highly processed foods and their correlation with diverse negative health consequences. Drawing from 45 different studies, the investigation unveiled clear links between the consumption of highly processed foods and several health metrics encompassing mortality, cancer, as well as mental, respiratory, cardiovascular, gastrointestinal, and metabolic health issues. Furthermore, following the predetermined criteria for evidence classification, compelling evidence substantiated direct links between a greater consumption of highly processed foods and elevated probabilities of cardiovascular disease-related mortality and the onset of type 2 diabetes.

Our mediation results are in line with existing findings on cancer.

Recent studies have demonstrated a significant association between the consumption of UPFs and the risk of overall and several cancers, including colorectal, breast, pancreatic [37], and ovarian cancer [38]. A still more recent study confirmed a positive association between UPFs intake and colorectal cancer risk [14], while no association was observed for colorectal and prostate cancers [32].

The mechanisms by which UPFs may influence the risk of chronic disease and multimorbidity are under study, but still not completely understood.

The high content of saturated fats and added sugars, increases the energy density of many UPFs (calories per weight or volume), in combination with a softer texture that leads to less chewing and delays the feeling of satiety, resulting in an effect on weight gain [39]. Obesity is an important risk factor and may initiate and promote progression to multimorbidity [40].

However, adjustment for BMI in our model did not negate the association with UPFs. Other studies indicate that the BMI-mediated effects are small and that other mechanisms are probably involved [41].

Diets with high UPFs were also associated with a lower nutritional quality, lower intake of dietary fiber and vitamins, and higher intake of free sugars and saturated fats [42]. 

However, in our cohort, all groups maintained an average adherence score to the Mediterranean diet.

Since adjustment for various dietary factors did not substantially attenuate the associations, it was hypothesized that ultra-processing itself may be associated with the disease risk, regardless of nutritional quality [43].

Salt and additives in ultra-processed foods may influence several physiological processes associated with cancer development, including inflammation, insulin resistance and alteration of gut flora [44].

Artificially sweetened UPFs (e.g., yoghurt, breakfast cereals, jelly desserts) may play a role in cancer incidence [45] because they contain potentially carcinogenic compounds such as aspartame and 4-methylimidazole [46] and contaminants from packaging materials (e.g., bisphenol A), newly formed contaminants produced during heat treatment, contaminants transferred from packaging materials and additives used to preserve and enhance the organoleptic properties of food [47].

Any of these can affect endocrine pathways or the gut microbiome and contribute to higher risk of chronic diseases [48,49]. 

This poses significant challenges for the food industry, suggesting that simply reformulating ultra-processed foods may not adequately address the risks associated with their consumption.

However, further research is needed to improve our understanding of these mechanisms and establish clearer causal relationships.

We recognize the limitations inherent in association studies: our results may fall into the domain of ‘significant but possibly misleading results’, where observed correlations may not result from causality, but emerge as a consequence of persistent confounding variables. However, the study has strengths and limitations that are useful to understand the results.

We employed a competitive risk assessment for the subsequent specific causes of death: cardiovascular disease, digestive disorders, cancer, and other causes. The competitive risk approach enables us to more effectively scrutinize the influence of various factors on the attainment of a composite end-point when the risk factor is connected to multiple outcomes of interest [50].

We lacked data regarding the physical activity of the participants, and the only proxy available for assessing physical activity was their job classification. Job categories like “craft, farming, and sales” and “elementary occupations” entail manual duties that result in greater energy expenditure, as opposed to other categories.

Another strength is that cancer cases were identified through registries that provide detailed information on cancer subtype, which are unlikely to be affected by measurement errors.

A limitation is that the FFQs used at baseline were not designed to distinguish between NOVA groups [50] and could lead to random misclassification and weakening of our association estimates. Residual confounding and the fact that the FFQs used were not designed to capture the extent and purpose of food processing may partly explain inconsistencies between studies. Other studies have expressed misgivings on this issue [41] 

In a recent study [51], the risks for individual food categories were assessed.

Among various categories of UPFs, heightened intake of artificially sweetened beverages and animal products exhibited a stronger correlation with increased multimorbidity risk, as also with increased consumption of sauces, spreads, and condiments, albeit with slightly less strength. Conversely, there was a slight inverse relationship between ultra-processed breads and cereals consumption and multimorbidity risk, though the certainty level was borderline. Sweets, desserts, savory snacks, plant-based alternatives, as well as ready-to-eat or reheated and mixed meals did not display any significant association with multimorbidity risk.

This approach could make the judgement of UPFs more flexible and perhaps produce practical indications for improving current nutritional guidelines. However, available evidence from prospective cohort studies suggests that adherence to a healthy dietary pattern such as the Mediterranean diet is associated with a reduced risk of several multimorbidity clusters [42]. 

Another limitation concerns a residual external comparability problem linked to the use of the categorization into quartiles of the UPF consumption variable. This was an obligatory choice in relation to the absence of consensual reference levels and the type of study linked to a specific cohort in Southern Italy. We believe that the threshold levels found in our study can fully enter into a subsequent discussion on the identification of national and international reference values.

As UPF consumption continues to increase in Mediterranean countries, public health interventions and dietary education programs are essential to mitigate the potential impact on public health and to promote the preservation of traditional Mediterranean dietary patterns.

## 5. Conclusions

In conclusion, our study provides further evidence of the link between ultra-processed foods and cancer risk, particularly gastrointestinal cancers, underlining the need for targeted interventions to improve diet and lifestyle with the aim of reducing the incidence of related diseases. Further research is needed to confirm and investigate these associations, as well as to identify effective preventive strategies at the individual and health policy levels. Meanwhile, a valid suggestion is to favor the consumption of fresh, unprocessed or minimally processed foods which should constitute the cornerstone of human nutrition

## Figures and Tables

**Figure 1 nutrients-16-01994-f001:**
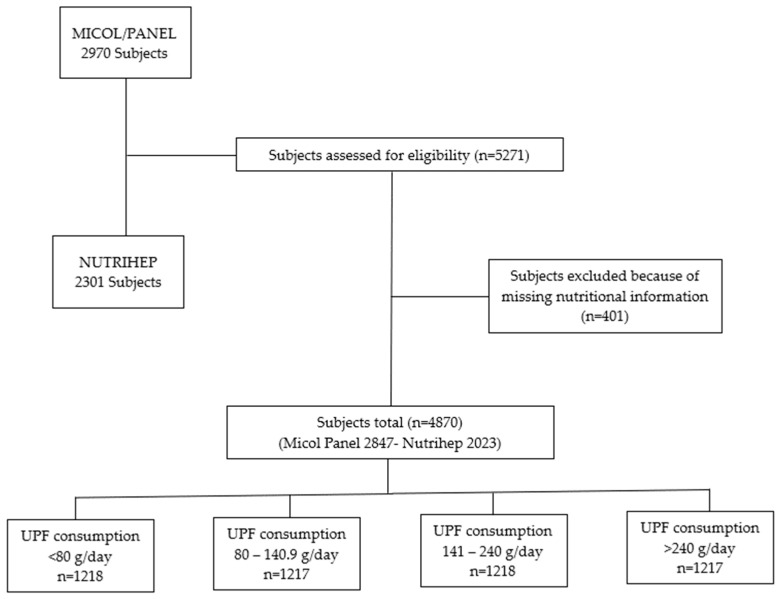
Flow Chart.

**Figure 2 nutrients-16-01994-f002:**
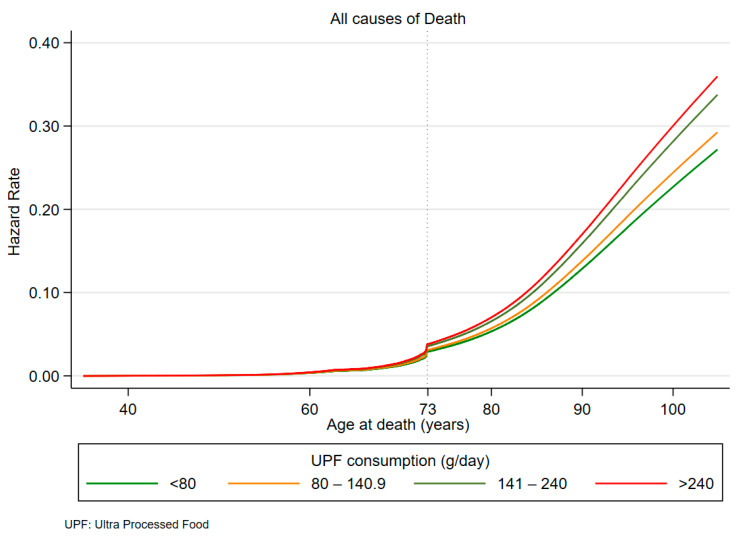
Hazard rates for all causes of death by UPF consumption quartiles.

**Figure 3 nutrients-16-01994-f003:**
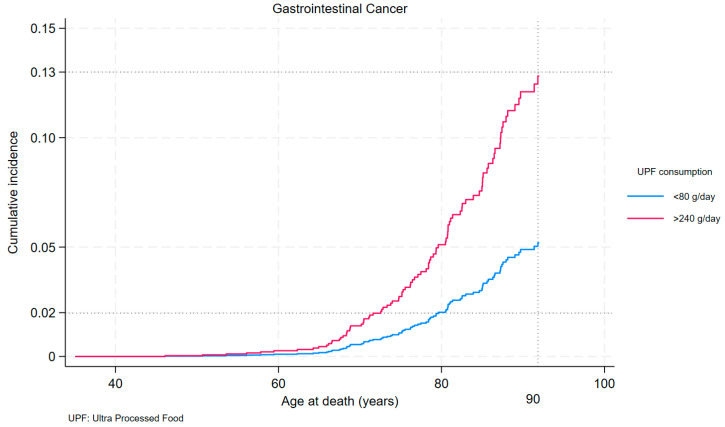
Cumulative incidence of death from Gastrointestinal Cancer by first and last quartiles of UPF consumption.

**Table 1 nutrients-16-01994-t001:** Characteristics of Participants by UPF consumption quartiles. MICOL/PANEL and NUTRIHEP Studies. Castellana Grotte. Putignano (BA). Italy. 2005–2023.

		Quartile of UPF Consumption (g/day)			
	All Sample	<80	80–140.9	141–240	>240
N	4870	1218	1217	1218	1217
Gender ^a^					
Female	2357 (48.4)	628 (26.6)	589 (25.0)	585 (24.8)	555 (23.5)
Male	2513 (51.6)	590 (23.5)	628 (25.0)	633 (25.2)	662 (26.3)
Enrollment Age ^b^ (years)	51.51 (15.84)	63.36 (12.79)	54.32 (13.73)	48.04 (13.54)	40.32 (13.53)
SBP ^b^ (mmHg)	124.00 (17.85)	130.33 (19.24)	124.34 (18.36)	122.25 (17.17)	119.06 (14.35)
DBP ^b^ (mmHg)	76.74 (9.75)	76.68 (10.02)	76.70 (9.87)	77.20 (9.70)	76.39 (9.39)
Weight ^b^ (kg)	73.18 (14.97)	73.06 (14.18)	73.76 (14.64)	73.41 (15.12)	72.50 (15.91)
BMI ^b^ (kg/m^2^)	27.58 (5.10)	29.10 (4.99)	28.02 (4.89)	27.16 (4.84)	26.05 (5.18)
BMI categories ^a^					
<25	1572 (32.3)	245 (15.6)	337 (21.4)	422 (26.8)	568 (36.1)
25–29.9	1888 (38.8)	507 (26.9)	506 (26.8)	464 (24.6)	411 (21.8)
30–40	1224 (25.1)	411 (33.6)	331 (27.0)	288 (23.5)	194 (15.8)
>40	184 (3.8)	54 (29.3)	43 (23.4)	44 (23.9)	43 (23.4)
Calories ^b^	2173.08 (794.13)	1762.38 (627.42)	1998.25 (643.51)	2199.41 (672.73)	2732.60 (866.35)
TG ^b^ (mg/dL)	122.06 (87.58)	134.54 (89.69)	125.91 (88.20)	113.98 (76.96)	113.82 (93.03)
TC ^b^ (mg/dL)	197.29 (39.19)	199.27 (39.31)	201.29 (39.28)	196.88 (37.93)	191.72 (39.63)
HDL-C ^b^ (mg/dL)	51.63 (13.63)	51.74 (14.07)	52.09 (14.02)	51.76 (13.49)	50.94 (12.90)
LDL-C ^b^ (mg/dL)	121.53 (33.65)	120.74 (34.39)	124.14 (33.35)	122.53 (33.14)	118.69 (33.51)
Glucose ^b^ (mg/dL)	105.87 (25.26)	114.10 (36.36)	107.02 (22.94)	102.53 (15.36)	99.84 (18.77)
ALT ^b^ (U/L)	16.70 (13.40)	17.83 (17.90)	16.53 (12.85)	16.23 (11.67)	16.23 (9.73)
Alcohol intake ^b^ (g/day)	20.60 (37.68)	21.21 (39.02)	23.02 (38.83)	21.89 (37.28)	16.27 (35.15)
Smoking habit ^a^					
Never	3950 (82.93)	1004 (25.4)	991 (25.1)	992 (25.1)	963 (24.4)
Current	813 (17.07)	165 (20.3)	200 (24.6)	206 (25.3)	242 (29.8)
rMED ^c^	8.00 (6.00–10.00)	9.00 (8.00–11.00)	8.00 (7.00–10.00)	8.00 (6.00–10.00)	7.00 (5.00–9.00)
Age at Death ^c^ (years)	68.33 (56.41–79.35)	80.06 (72.31–86.71)	71.82 (60.76–80.33)	63.65 (55.29–74.08)	56.01 (48.35–65.53)
Observation Time ^c^ (years)	17.82 (17.07–18.05)	17.78 (13.19–18.28)	17.87 (17.07–18.20)	17.82 (17.15–17.95)	17.77 (17.20–17.92)
Status ^a^					
Alive	3935 (80.8)	747 (19.0)	979 (24.9)	1065 (27.1)	1144 (29.1)
Dead	935 (19.2)	471 (50.4)	238 (25.5)	153 (16.4)	73 (7.8)
Cause of Death ^a^					
CVD	271 (29.0)	150 (55.4)	62 (22.9)	38 (14.0)	21 (7.7)
GC	105 (11.2)	43 (41.0)	34 (32.4)	13 (12.4)	15 (14.3)
OCr	163 (17.4)	62 (38.0)	51 (31.3)	39 (23.9)	11 (6.7)
DOC	396 (42.3)	216 (54.5)	91 (23.0)	63 (15.9)	26 (6.6)
Job ^a^					
Jobless and Pensioners	1643 (33.9)	601 (36.6)	433 (26.4)	330 (20.1)	279 (17.0)
Managers and Professionals	285 (5.9)	27 (9.5)	61 (21.4)	79 (27.7)	118 (41.4)
Craft, AGricultural and Sales Workers	1218 (25.1)	216 (17.7)	288 (23.6)	335 (27.5)	379 (31.1)
Housewife	569 (11.7)	139 (24.4)	149 (26.2)	160 (28.1)	121 (21.3)
Elementary Occupations	1133 (23.4)	231 (20.4)	282 (24.9)	307 (27.1)	313 (27.6)
Education^a^					
Illiterate	167 (3.46)	57 (34.1)	65 (38.9)	30 (18.0)	15 (9.0)
Primary School	1328 (27.5)	481 (36.2)	358 (27.0)	294 (22.1)	195 (14.7)
Secondary School	1475 (30.5)	373 (25.3)	366 (24.8)	357 (24.2)	379 (25.7)
High School	1374 (28.4)	208 (15.1)	303 (22.1)	373 (27.1)	490 (35.7)
Graduated	488 (10.1)	88 (18.0)	120 (24.6)	154 (31.6)	126 (25.8)
Marital Status ^a^					
Single	756 (15.6)	107 (14.2)	95 (12.6)	178 (23.5)	376 (49.7)
Married or Cohabiting	3649 (75.5)	899 (24.6)	1.024 (28.1)	948 (26.0)	778 (21.3)
Separated or Divorced	117 (2.42)	29 (24.8)	30 (25.6)	26 (22.2)	32 (27.4)
Widower	310 (6.42)	172 (55.5)	63 (20.3)	56 (18.1)	19 (6.1)
Hypertension ^a^					
No	3665 (75.3)	685 (18.7)	880 (24.0)	1006 (27.4)	1094 (29.8)
Yes	1205 (24.7)	533 (44.2)	337 (28.0)	212 (17.6)	123 (10.2)
Dyslipidemia ^a^					
No	4063 (83.4)	880 (21.7)	972 (23.9)	1074 (26.4)	1137 (28.0)
Yes	807 (16.6)	338 (41.9)	245 (30.4)	144 (17.8)	80 (9.9)
Diabetes ^a^					
No	4544 (93.3)	1027 (22.6)	1149 (25.3)	1179 (25.9)	1189 (26.2)
Yes	326 (6.70)	191 (58.6)	68 (20.9)	39 (12.0)	28 (8.6)

UPFs: Ultra Processed Foods; SBP: Systolic Blood Pressure; DBP: Diastolic Blood Pressure; BMI: Body Mass Index; TG: Triglycerides; TC: Total Cholesterol; HDL-C: High-Density Lipoprotein Cholesterol; LDL-C: Low-Density Lipoprotein Cholesterol; ALT: Alanine Aminotransferase; CVD-related mortality: Cardiovascular Disease-related mortality; GC: Gastrointestinal Cancer; OCr: Other Cancers; and DOC: Deaths from Other Causes. ^a^ Number (Percentage), ^b^ Mean ± (SD), and ^c^ Median (IQR). Percentages calculated for the column.

**Table 2 nutrients-16-01994-t002:** Mortality Hazard Ratios (HR) and Subdistribution Hazard Ratios (SHR) according to the UPF consumption quartiles.

	All Causes	CVD	GC	OCr	DOC
	HR (95% CI)	SHR (95% CI)	SHR (95% CI)	SHR (95% CI)	SHR (95% CI)
Model 1					
Q1	referent	referent	referent	referent	referent
Q2	1.10 (0.93 1.29)	0.90 (0.66; 1.23)	1.50 (0.94; 2.39)	1.44 (0.98; 2.10)	0.94 (0.72; 1.21)
Q3	1.26 * (1.03; 1.53)	1.03 0.72; 1.49)	0.91 (0.47; 1.74)	1.73 * (1.15; 2.62)	1.13 (0.84; 1.51)
Q4	1.32 * (1.02; 1.72)	1.19 (0.73; 1.92)	2.43 * (1.33; 4.45)	0.96 (0.50; 1.85)	1.10 (0.73; 1.67)
UPFs continuous	1.00 * (1.0001; 1.001)	1.00 (0.999; 1.001)	1.00 * (1.0002; 1.002)	1.00 (0.999; 1.001)	1.00 (1.000; 1.001)
Model 2					
Q1	referent	referent	referent	referent	referent
Q2	1.09 (0.92; 1.30)	0.84 (0.60; 1.16)	1.65 * (1.01; 2.71)	1.46 (0.98; 2.17)	0.89 (0.68; 1.17)
Q3	1.27 * (1.03; 1.57)	1.00 (0.68; 1.47)	0.96 (0.47; 1.93)	1.61 * (1.03; 2.54)	1.03 (0.75; 1.42)
Q4	1.34 * (1.00; 1.79)	1.33 (0.76; 2.30)	3.14 ** (1.56; 6.32)	1.11 (0.55; 2.24)	1.14 (0.71; 1.82)
UPFs continuous	1.00 * (1.000; 1.001)	1.00 (0.999; 1.002)	1.00 * (1.000; 1.002)	1.00 (0.999; 1.002)	0.999 (0.998; 1.001)

* *p*-value < 0.05 ** *p*-value <0.001. Model 1: including only UPF consumption quartiles; Model 2: adjusted for sex (F vs. M), BMI categories, Alanine Aminotransferase, Triglycerides, Glucose, Smoke, Job, Marital Status, Alcohol intake, and Daily Calories. HR: Hazard Ratio; SHR: Subdistribution Hazard Ratio; UPFs: Ultra Processed Food; CVD: Cardiovascular Disease; GC: Gastrointestinal Cancer; OCr: Other Cancers; and DOC: Deaths from Other Causes.

## Data Availability

The original contributions submitted in the study are incorporated in the article. Additional enquiries may be addressed to the corresponding author.

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
