# Peer review of "Ultra-Processed Food Consumption as a Risk Factor for Gastrointestinal Cancer and Other Causes of Mortality in Southern Italy: A Competing Risk Approach"

_nutrients, 2024, doi:10.3390/nu16131994_

Round 1

Reviewer 1 Report

Comments and Suggestions for Authors

Great job with the piece. The only thing that I noted for you to offer more commentary on in the Limitations is the use of quartiles to divide the groups. You mentioned the exercise component in the Limitations....good job. But, you do need to consider the quartile issues and comparability.

Author Response

The choice to use quartiles arose from the absence of guidelines on the "optimal" consumption of UPFs and universally accepted risk thresholds. Many articles in the literature relating to UPFs use the exposure variable categorized into quartiles or quintiles, perhaps also because the propensity to consume UPFs is variable in different populations (we report some articles, also cited in the bibliography, on studies like ours which use quartiles or quintiles). In any case, we are aware that the use of quartiles may introduce a problem of external comparability and we have decided to include a sentence in this regard in the paragraph on limitations (line 331).

Chang K, Gunter MJ, Rauber F, Levy RB, Huybrechts I, Kliemann N, Millett C, Vamos EP.

Ultra-processed food consumption, cancer risk and cancer mortality: a large-scale prospective analysis within the UK Biobank. EClinicalMedicine. 2023 Jan 31;56:101840

Fang Z, Rossato SL, Hang D, Khandpur N, Wang K, Lo CH, Willett WC, Giovannucci EL, Song M.

Association of ultra-processed food consumption with all-cause and cause-specific mortality: population based cohort study. BMJ. 2024 May 8;385

Juul F, Vaidean G, Lin Y, Deierlein AL, Parekh N. Ultra-Processed Foods and Incident Cardiovascular Disease in the Framingham Offspring Study. J Am Coll Cardiol. 2021 Mar 30;77(12):1520-1531

Reviewer 2 Report

Comments and Suggestions for Authors

This paper is well written and conclusions is satisfactorily introduced.

However, in results section death rate is quite high because of patients' age have been increased during long-term follow-up period (more than 17 years).

There is one question about death-age tendency (Figure 1 and 2); is this tendency consistent with Nothern part of Italy where much more richer people is living? Answer to this question shoud be added in discussion.

---

Please consider some specific comments from reviewers such as:

1. What is the main question addressed by the research?

Reviewer's comment: This review aimed at how to select biologics and small molecules based on the first class evidence.

2. What parts do you consider original or relevant for the field? What 

specific gap in the field does the paper address? 

Reviewer's comment: This paper is aiming at people living in their country and there  are a lot of differences from guidelines in the Western developed countries. This difference might be accepted because of socioeconomic circumstances in each country.

3. What does it add to the subject area compared with other published 

material?

Reviewer's comment: Evidence quoted in this review seems to be appropriate from the methodological standpoint.

4. What specific improvements should the authors consider regarding the methodology? What further controls should be considered?

Reviewer's comment: The conclusions are consistent with the other guidelines. However, conclusions have to be presented as a flowchart which shows  the first line selection, second line selection and maintenance phase treatment. In addition, when loss of response occurred how to switch the next step selection.

5. Are the references appropriate?

Reviewer's comment: The references quoted are appropriate.

Author Response

Dear Reviewer, 

1) However, in results section death rate is quite high because of patients' age have been increased during long-term follow-up period (more than 17 years).

Certainly the factor most linked to the death rate is the attained age, as we explained in the "statistical analysis" section (line 141), however, having used age as a temporal axis, we believe we have addressed this issue, allowing comparability between the consumption quartiles of UPFs. Further inclusion of age as a covariate in the model would have been redundant and would have compromised interpretability.

2) There is one question about death-age tendency (Figure 1 and 2); is this tendency consistent with Nothern part of Italy where much more richer people is living? Answer to this question shoud be added in discussion.

We are unable to give a detailed answer. Ours is a population study on a specific cohort and we have described our reality. We did not find any studies in the literature that described the situation in Northern Italy with the same outcomes and risk factors. Of course, the economic status influences both the type of consumption and the risk of death, which is why we used the job status as a correction variable. 

Reviewer 3 Report

Comments and Suggestions for Authors

Dear Editor, dear Authors,

The publication reviewed here can be thoroughly recommended for acceptance. The methodological weaknesses of the work are clearly addressed and named in the discussion. A flow chart takes up the recruitment and the introduction leads to the actual research question and provides a good introduction to the topic. All individual steps are methodologically and statistically comprehensible.

Author Response

Dear Reviewer,

thank you very much. It was a pleasure to receive your review